# Tissue localization of natural killer cells dictates surveillance of lung metastasis

Marijne Vermeer [1], Colin Sparano[1], Nicolò Coianiz [1], Maud Mayoux [1], Gioana Litscher[1], André Fonseca[2], Stanislav Dergun [1], Caroline Mussak[1], Lukas Rindlisbacher[2], Philipp Häne [2], Laura Ducimetière[1], Tobias Wertheimer[3], Burkhard Becher [2], Lothar C. Dieterich [4] & Sònia Tugues [1,5] ✉

The lung is a common metastatic site for various cancer types. Successful immune surveillance of lung metastasis depends on Natural Killer (NK) cells, but the underlying mechanisms are elusive. Here, we show that the pulmonary vasculature recruits and maintains highly cytotoxic, differentiated CD11b^high NK cells through the integrins Lymphocyte Function-associated Antigen 1 (LFA-1) and Very Late Antigen (VLA-4). These NK cells rapidly eradicate metastasizing tumor cells within the vasculature. However, after the initial clearing phase, differentiated pulmonary NK cells largely remain intravascular and fail to track extravasated tumor cells. In contrast, metastatic nodules are preferentially infiltrated with circulating, less differentiated CD27^high NK cells. Within the metastatic lung, CD11b^high NK cells undergo a rapid impairment of their migratory and cytotoxic features, while the intranodular CD27^high subset transitions towards a transforming growth factor β (TGF-β)-driven state with limited persistence. Our findings demonstrate that the compartmentalization of NK cells is key for effective tumor cell surveillance in lung metastasis and suggests that TGF-β-resistant CD27^high NK cells may offer a promising therapeutic avenue to enhance local anti-tumor activity.

The lung is a common site for metastasis from various cancer types, including breast, skin and colon cancer[1,2]. In fact, about 20 to 50% of cancer-related deaths involve pulmonary metastases, underscoring the need to understand the mechanisms governing the metastatic process in this organ[3,4]. Metastasis relies on the ability of tumor cells to invade surrounding tissues, enter the circulation and extravasate to distal organs, resulting in the formation of secondary tumors[5,6]. Through all these stages, disseminating tumor cells establish fine-tuned interactions with the host immune system, which either promote or suppress metastatic outgrowth.

A key player in the lung's immune response against metastases is the Natural Killer (NK) cell, an innate lymphocyte with high cytotoxic potential[7]. The activity of NK cells is tightly regulated by inhibitory and activating receptors, which facilitate rapid and specific responses to environmental cues[8,9]. Upon activation, NK cells release IFN-γ or induce target cell death through the secretion of cytotoxic granules, containing granzyme B (Gzmb) and perforin, or via death-receptor pathways[10,11]. NK cells are highly abundant in the lung, accounting for 10–20% of the total lymphocyte population[12] and have been widely reported to clear metastasizing tumor cells in this organ[13–19]. However,

[1]Innate Lymphoid Cells and Cancer, Institute of Experimental Immunology, University of Zurich, Zurich, Switzerland. [2]Inflammation Research, Institute of Experimental Immunology, University of Zurich, Zurich, Switzerland. [3]Department of Internal Medicine I (Hematology, Oncology and Stem Cell Transplantation), University Medical Center Freiburg, Freiburg im Breisgau, Germany. [4]European Center for Angioscience (ECAS), Medical Faculty Mannheim, Heidelberg University, Mannheim, Germany. [5]Department of Immunology, Medical Faculty Mannheim, Mannheim Institute for Innate Immunosciences (MI3), Heidelberg University, Mannheim, Germany. ✉e-mail: tugues@immunology.uzh.ch

during the progression of lung metastasis, pulmonary NK cells become functionally impaired and lose the ability to control metastatic outgrowth[18,20,21]. While this can be partially attributed to the ability of cancer cells to cleave ligands for activating receptors[22] or to release immunosuppressive factors (e.g. IL-10, TGF-β or PGE2)[8,23], the dichotomy between a highly efficient initial tumor cell elimination and the subsequent failure to control outgrowth remains puzzling.

Pulmonary NK cells are highly differentiated and characterized by CD11b[high] and CD56[dim] phenotypes in mice and in humans respectively[12,24,25]. In mice, the CD11b[high] NK cell subset represents the end stage of a maturation cascade that starts with bone marrow-derived CD11b[low]CD27[low] NK cells, which differentiate through CD11b[low]CD27[high] and CD11b[high]CD27[high] stages towards CD11b[high]CD27[low] NK cells[24]. Throughout this maturation process, NK cells modulate their expression of activating and inhibitory receptors, adhesion molecules and chemokine receptors, concomitant with the acquisition of functional proficiency by expressing cytotoxic molecules and cytokines[26]. The more differentiated CD11b[high]CD27[low] NK cell subset, for instance, features high levels of the C-type lectin inhibitory receptor KLRG-1, Gzmb and the chemokine receptor CX3CR1. In contrast, CD27[high] NK cells constitutively express CXCR3 and produce large amounts of cytokines upon stimulation[24,26–28]. Traditionally, tumor growth has been associated with an impaired functional maturation of NK cells[29]. However, the specific contribution of various maturation stages of NK cells to metastatic surveillance remains unclear.

Here, we reveal a spatial and functional compartmentalisation of NK cells in surveilling pulmonary metastasis. Differentiated CD11b[high] NK cells that are patrol the lung vasculature and rapidly eliminate circulating tumor cells, yet remain intravascular and fail to track extravasating tumor cells. In contrast, CD27[high] NK cells infiltrate metastatic nodules more efficiently but adopt a TGFβ-driven program that limits their persistence. Together, our highlight an unanticipated division of labor of NK cells relevant for anti-metastatic therapies.

## Results

### NK cells rapidly clear lung-metastasizing tumor cells inside the vasculature

Despite the well-established role of NK cells in eliminating metastatic tumor cells in the lung, the kinetics of this process are still poorly understood. To tackle this question, we used a model of experimental lung metastasis based on the intravenous (i.v) injection of luciferase (luc)[+] PyMT-B6, a cell line derived from the MMTV-Polyomavirus Middle T (PyMT) mammary tumor model on the C57BL/6 background[30], in which metastases occur almost exclusively in the lung. To study the critical timepoint for NK cell activity during this metastatic process, we used an anti-NK1.1 antibody to deplete NK cells either 24 h before, 24 h after or 7 days after tumor cell injection (Fig. 1A). We observed an increased metastatic burden only when NK cells were depleted before tumor cell inoculation and not thereafter, confirming the previously described role of NK cells in metastatic seeding but not metastatic outgrowth (Fig. 1B)[30,31]. To assess whether this timing holds true in a more clinically relevant setting, we turned to the spontaneous PyMT model with surgical resection of the primary tumor. In this context, NK cell depletion either before or after surgery revealed increased metastatic burden only when NK cells were depleted prior to resection, despite no observable change in primary tumor growth before surgery (Figure S1A–D), also suggesting that NK cells limit dissemination at early stages. To further investigate the dynamics of early tumor cell elimination in the lung, we assessed the clearance kinetics within the very first hours upon luc[+]PyMT-B6 tumor cell injection (Fig. 1C). We found that tumor cells metastasizing to the lung were rapidly cleared, as shown by the decreased luciferase signal already 6 h after tumor cell inoculation (Fig. 1D). Importantly, this process was dependent on NK cells, since NK cell depletion prevented lung metastatic clearance at the time points analyzed (Fig. 1D). We corroborated the rapid tumor

clearance by NK cells using the melanoma cell line B16-F10 (Figure S1E, F). Together, these findings highlight a crucial role of NK cells in the elimination of metastasizing tumor cells upon arrival in the lung.

To better understand where the metastatic clearance takes place, we investigated the location of NK cells in the lung. The lung is a highly vascularized organ and, in steady-state conditions, many leukocytes localize in the vascular compartment of this tissue[32]. We therefore injected fluorescently labeled anti-CD45 antibody i.v. followed by a second anti-CD45 antibody ex vivo to distinguish intravascular (double positive) and extravascular (only ex vivo positive) leukocytes[32,33]. In steady-state conditions, we found that nearly all NK cells were located intravascularly (Fig. 1E, F and Figure S1G, H), in contrast to the full or partial extravascular localization of other immune cells (e.g., alveolar macrophages, dendritic cells) (Figure S1I-J). We confirmed the intravascular localization of lung NK cells by immunofluorescence, using Ncr1[Cre]R26R[Ai14] reporter mice to visualize NK cells and an antibody against VE-cadherin to stain the pulmonary endothelium (Fig. 1H). Interestingly, NK cells remained associated to the vasculature upon inoculation of PyMT-B6 tumor cells and did not abandon this anatomical location during the timeframe of metastatic clearance (Fig. 1G and Figure S1G). Also in this setting, we confirmed the intravascular localization of NK cells in the lung by immunofluorescence (Fig. 1I). Overall, our data reveal that NK cells are associated to the pulmonary vasculature, where they remain during early metastatic clearance and effectively eliminate tumor cells.

### Differentiated CD11bhigh NK cells are preferentially recruited to the lung

We next sought to better characterize the phenotypical features of pulmonary intravascular NK cells by flow cytometry. We found that intravascular lung NK cells display a more differentiated phenotype (CD11b[high]CD27[low]) in comparison to blood or splenic NK cells, which feature more of the less differentiated CD27[high] subsets (Fig. 2A, B). In agreement with previous reports[24], differentiated lung NK cells express high levels of KLRG1, GzmB and CX3CR1, but low levels of CXCR3 (Fig. 2C, D). The highly differentiated phenotype of pulmonary NK cells led us to hypothesize that the lung vasculature either specifically attracts differentiated CD11b[high] NK cells or facilitates the differentiation of CD27[high] NK cells intravascularly. To distinguish between these two hypotheses, we transferred CD27[high] NK cells from CD45.1 mice together with the CD11b[high] NK cells from CD45.2 mice at a 1:1 ratio into lymphopenic Rag2[-/-]Il2rg[-/-] mice and investigated their distribution across different organs 12 h and 72 h after transfer (Fig. 2E). Strikingly, within the 12 h timeframe, the lung vasculature was preferentially occupied by differentiated CD11b[high] NK cells, while blood and spleen maintained a 1:1 ratio of CD11b[high] and CD27[high] NK cells (Fig. 2F, G). We did not observe a differentiation of the CD27[high] subset into CD11b[high] NK cells within this short time interval (Fig. 2H), ruling out the possibility that the increased percentages of CD11b[high] NK cells in the lung result from in situ differentiation. At 72 h after adoptive transfer, CD27[high] NK cells had differentiated into the CD11b[high] subset (Fig. 2I, J). Moreover, not only lung NK cells but also splenic NK cells had differentiated into CD11b[high] NK cells (Fig. 2I, J), supporting the concept that differentiated NK cells exhibit tropism for the lung, rather than the lung being a preferential site for NK cell maturation.

Differentiated NK cells express CX3CR1 (Fig. 2C, D)[28], which mediates cell adhesion and trafficking through its interactions with the chemokine CX3CL1 (fractalkine)[34]. By leveraging two publicly accessible single cell RNA sequencing (scRNA-seq) datasets[35,36] we found that, even under steady-state conditions, the pulmonary endothelium features high expression of CX3CL1 in comparison to other organs (Fig. S2A–C), particularly in capillaries (Fig. S2D–F). We therefore investigated whether the enhanced recruitment of differentiated NK cells to the lung relies on the CX3CL1-CX3CR1 signaling axis. To test this, we used CX3CR1-GFP knock-in/knock-out mice, which lack the

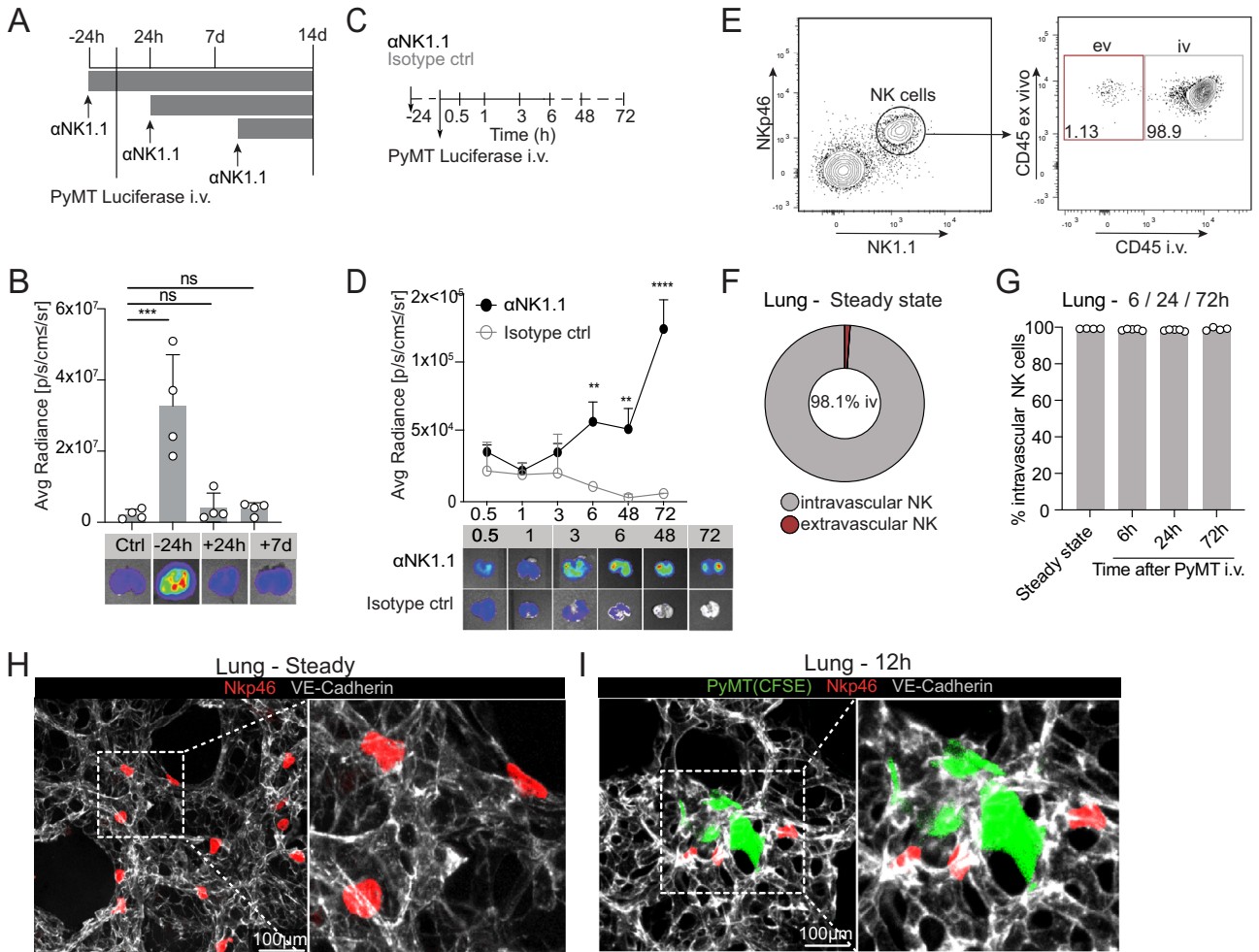

**Fig. 1 | Intravascular NK cells rapidly clear metastasizing tumor cells to the lung. A, B** PyMT Luciferase cells ($5 \times 10^5$) were injected intravenously (i.v.) to examine metastatic cell clearance. NK cells were depleted using 200 µg of anti-NK1.1 antibody 24 h before, 24 h after or 7 days after tumor cell inoculation. At 14 days post-injection, metastatic load in the lungs was measured using ex vivo In Vivo Imaging Systems (IVIS). Bioluminescent measurements on the whole lungs are shown. **A** Schematic illustration of the experimental approach. **B** Bar chart and corresponding representative IVIS images showing PyMT lung metastasis by IVIS at the endpoint. **C, D** Early tumor cell clearance in the lungs of mice injected i.v. with PyMT Luciferase cells ($5 \times 10^5$) from 0.5 h until 72 h after tumor injection. NK cells were depleted using 200 µg of anti-NK1.1 antibody 24 h prior to tumor injection and tumor cells were quantified using IVIS. **C** Schematic illustration of experimental approach. **D** Kinetics of metastatic load in the lungs measured by IVIS with corresponding representative IVIS images. **E** Representative contour plots depicting the percentages of intravascular (CD45iv$^+$, iv) and extravascular (CD45iv$^-$, ev) NK cell populations in the lung. Samples were pre-gated on single live CD45$^+$lineage$^-$ cells and subsequently gated on NK1.1$^+$NKp46$^+$ cells. **F** Donut plot indicating the percentages of intravascular and extravascular NK cells in the lung. **G** Bar chart showing the frequency of CD45iv$^+$ intravascular NK cells within the lungs of mice injected with PyMT tumor cells i.v. Tumor cells were injected 6 h, 24 h and 72 h before the endpoint. **H** Immunofluorescence staining of the lung of Ncr1$^{iCre}$Ai14$^{fl/wt}$ mice showing NKp46$^+$ cells (red) within the vasculature stained with VE-cadherin (gray). Scale bar: 100 µm. **I** Immunofluorescence staining of the lung of Ncr1$^{iCre}$Ai14$^{fl/wt}$ mice injected with CSFE-labeled PyMT cells (green) 12 h before sacrifice with NKp46$^+$ cells (red) within the vasculature (VE-cadherin/gray). Scale bar: 50 µm. **A–D** Data representative for 3 independent experiments with $n = 3$–4 mice or (**G**) 2 independent experiments with $n = 4$–5 mice. Error bars display means ± SD. Statistical significance was calculated using one-way analysis of variance (ANOVA) with Tukey's multiple comparisons test or a Two-way ANOVA with a Sidak's post hoc test; *$P < 0.05$, **$P < 0.01$, ***$P < 0.001$, and ****$P < 0.0001$. ns, not significant.

CX3CR1 receptor due to homozygous deletion[37]. The transfer of CD11b$^{high}$ differentiated NK cells from either WT (CD45.1) or CX3CR1$^{GFP/GFP}$ (CD45.2) mice into Rag2$^{-/-}$Il2rg$^{-/-}$ mice indeed revealed a decrease in homing towards the lungs by CX3CR1-deficient NK cells (Fig. S2G, H), although additional mechanisms may be required for the tropism of differentiated NK cell to this tissue. In summary, these findings highlight a selective recruitment of differentiated NK cells to the lung vasculature.

**NK cells anchor to the pulmonary vasculature through LFA-1 and VLA-4 which enables metastatic surveillance**

Our data showed that differentiated NK cells localise and clear tumor cells within the lung vasculature. We thus sought to better understand how NK cells interact with the pulmonary endothelium in the context of metastatic clearance. To analyze the crosstalk between NK cells and lung endothelial cells we used CellChat[38] on scRNA-seq datasets of lung NK cells and pulmonary endothelium[36]. CellChat predicted strong interactions between NK cells and lung endothelial cells (divided in artery, capillary 1, capillary 2, vein and lymphatic endothelium), with integrin-ligand pairs as main contributors of this crosstalk (Figs. 3A, and Fig. S3A). Specifically, we identified *ITGAL/ITGB2* (encoding for CD11a/CD18 or LFA-1) on NK cells interacting with Intercellular adhesion molecule 1 (*ICAM1*) and *ICAM2*, expressed on different subsets of endothelial cells, as well as *ITGA4/ITGB1* (encoding for CD49d/CD29 or VLA-4) on NK cells binding to Vasculature cell adhesion molecule 1 (*VCAM1*) on veins (Fig. 3A). Integrins such as LFA-1 and VLA-4 enable leukocytes to stably adhere to the endothelium through interactions with ICAM-1 and VCAM-1, respectively[39,40]. Additional integrins

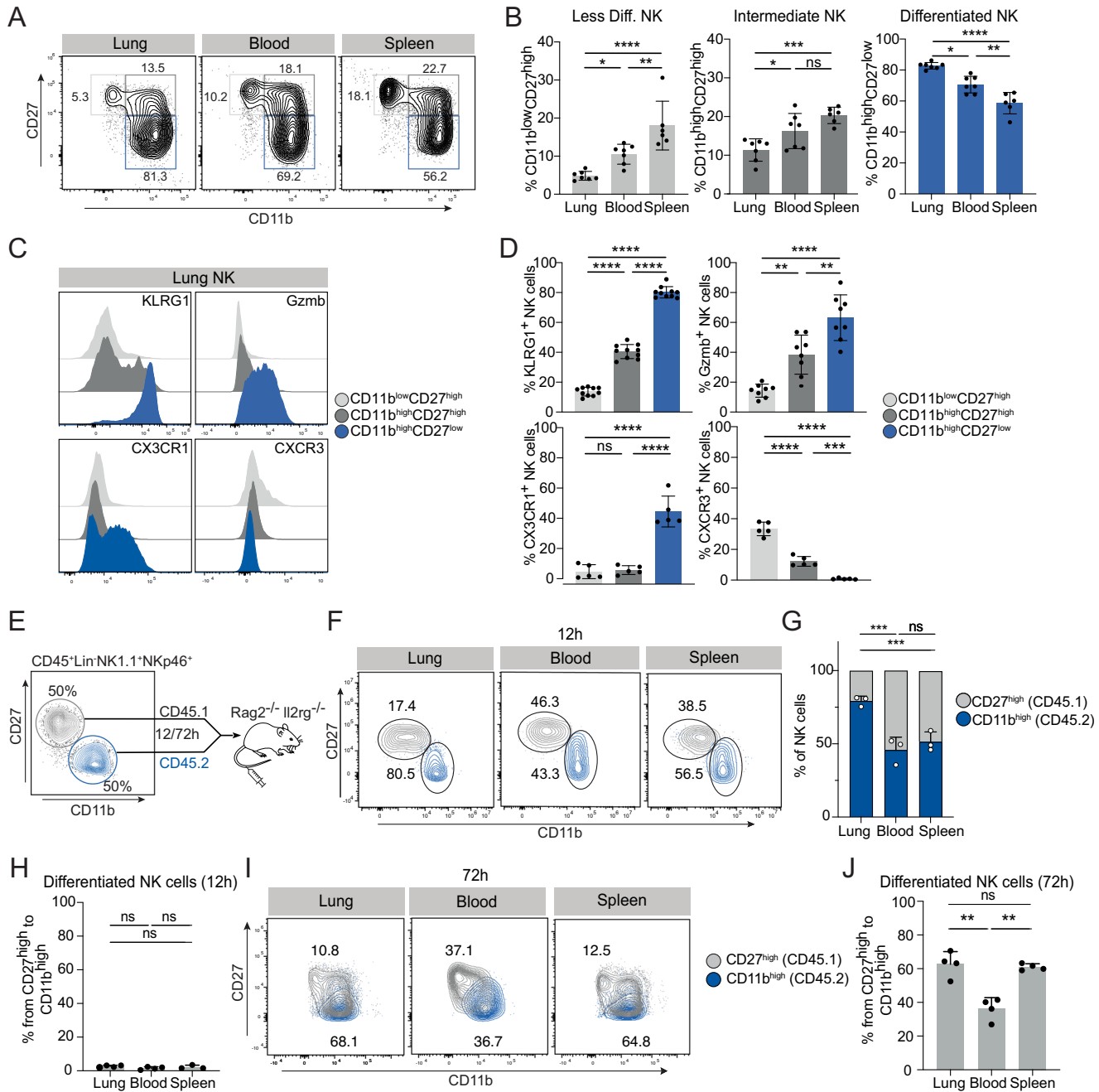

**Fig. 2 | Differentiated NK cells are preferentially recruited to the lung.**
**A** Representative contour plots depicting differentiation stages of NK cells in the lung, blood and spleen. NK1.1+NKp46+ NK cells were gated as CD11blowCD27high, CD11bhighCD27high and CD11bhighCD27low cells. **B** Bar charts displaying the frequency of NK cell subsets (CD11blowCD27high, CD11bhighCD27high and CD11bhighCD27low, subsequently) of total NK cells within the lung, blood and spleen. **C** Representative histograms showing the expression of KLRG1, Granzyme B (Gzmb), CX3CR1, CXCR3 on CD11blowCD27high, CD11bhighCD27high and CD11bhighCD27low NK cells in the lung. **D** Bar charts displaying the frequency of KLRG1+, Gzmb+, CX3CR1+ and CXCR3+ NK cells in the lung. **E–J** FACS-sorted CD45.1+CD27high (gray) and CD45.2+CD11bhigh (blue) NK cells were adoptively transferred at a 1 to 1 ratio in *Rag2−/−Il2rg−/−* mice and their distribution was analyzed 12 h and 72 h post-transfer. **E** Schematic illustration of experimental approach. **F** Contour plots depicting the transferred

CD45.1+CD27high and CD45.2+CD11bhigh NK cells 12 h post-transfer in the lung, blood and spleen of *Rag2−/−Il2rg−/−* mice. **G** Bar chart showing the ratio of CD45.1+CD27high NK cells and CD45.2+CD11bhigh NK cells in the lung, blood and spleen. **H** Bar chart presenting the conversion rate of CD45.1+CD27high NK cells to CD11bhigh NK cells within a 12 h timeframe post-transfer. **I** Contour plots depicting the transferred CD45.1+CD27high (gray) and CD45.2+CD11bhigh (blue) NK cells 72 h post-transfer in the lung, blood and spleen of *Rag2−/−Il2rg−/−* mice. **J** Bar chart presenting the conversion rate of CD45.1+CD27high NK cells to CD11bhigh NK cells within a 72 h timeframe post-transfer. Data is (**A–D**) pooled from 2–3 experiments with n = 2–4 or (**F–J**) representative from 2 experiments with n = 3–4. Error bars display means ± SD. Statistical significance was determined using (**B, D, H**) One-way ANOVA with Tukey's multiple comparisons test or (**G**) a Two-way ANOVA with a Sidak's post hoc test; *P < 0.05, **P < 0.01, ***P < 0.001, and ****P < 0.0001. ns, not significant.

predicted to mediate interactions between NK cells and the pulmonary endothelium include *ITGAM/ITGB2* (encoding for CD11b/CD18 or Mac-1) and *ITGA4/ITGB7*, which interact with ICAM-1/ICAM-2 and VCAM-1, respectively (Fig. 3A and Figure S3A). In line with high mRNA

expression of *Itgal* (CD11a), *Itgb2* (CD18), *Itga4* (CD49d) and *Itgb1* (CD29) on lung NK cells (Fig. 3B), we found these integrins present at the protein level (Fig. 3C). Similarly, we verified high levels of *Icam1*, *Icam2* and *Vcam1* in steady-state pulmonary endothelium (Fig. 3B) by

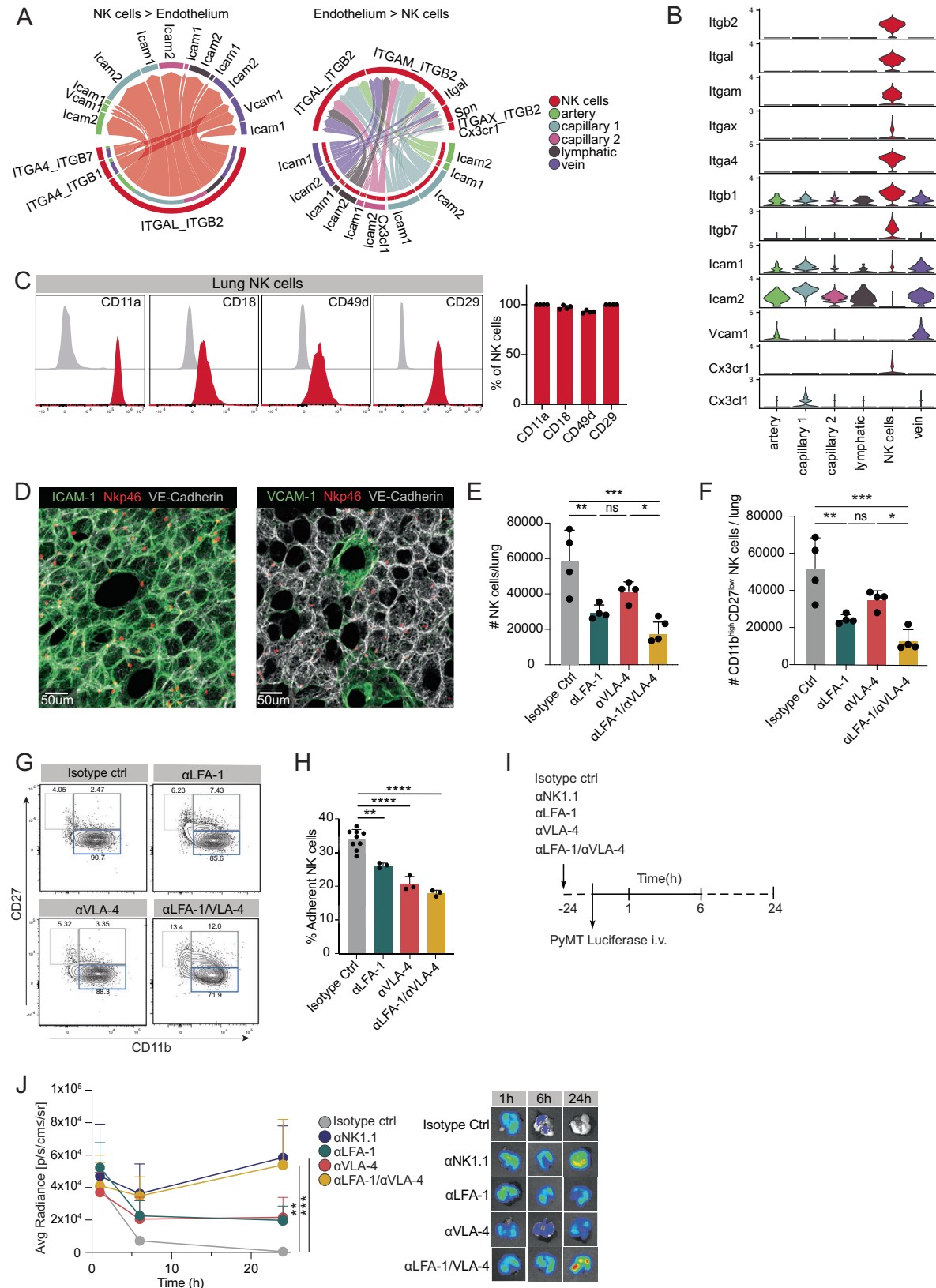

immunofluorescence in lungs of $Ncr1^{Cre}R26R^{Ai14}$ reporter mice (Fig. 3D). Both ICAM-1 and VCAM-1 were co-expressed with the vessel marker VE-cadherin along the pulmonary endothelium (Fig. 3D).

Given that both LFA-1/ICAM-1 and VLA-4/VCAM-1 pairs were predicted to facilitate NK cell crosstalk with the pulmonary endothelium, we sought to explore the impact on pulmonary NK cells when

disrupting these interactions. We therefore used anti-LFA-1 and anti-VLA-4 blocking antibodies and characterized NK cells from lungs in steady-state conditions using flow cytometry. As shown in Fig. S3B, both CD11a and CD49d on lung NK cells were efficiently blocked by the anti-LFA-1 and anti-VLA-4 antibodies, respectively. Inhibition of LFA-1 and VLA-4 led to decreased numbers of intravascular lung NK cells

**Fig. 3 | Lung metastatic cell clearance depends on the integrins LFA-1 and VLA-4. A, B** Analysis of scRNAseq datasets of murine pulmonary endothelium[36] and murine pulmonary NK cells using CellChat[38]. **A** Chord diagrams depicting the significant integrin signaling pathways from NK cells to different pulmonary endothelial clusters (left) and from endothelial cell clusters to NK cells (right). Width of arrows is proportional to the interaction strength between ligand-receptor pairs. **B** Violin plots showing the differential expression of various genes in NK cells and the pulmonary endothelial clusters. **C** Representative histograms and bar chart showing expression of CD11a, CD18, CD49d and CD29 in NK1.1⁺NKp46⁺ pulmonary NK cells. Data are representative for one of 2 independent experiments with $n = 4$ mice per group. **D** Immunofluorescence stainings of the lung of $Ncr1^{iCre}Ai14^{fl/wt}$ mice with NKp46⁺ cells (red) and VE-cadherin endothelium (gray) showing ICAM-1 (left, green) and VCAM-1 expression (right, green). Scale bar: 50 μm. **E–G** anti-LFA-1, anti-VLA-4, anti-LFA-1 and anti-VLA-4 or isotype control antibody were intravenously administered, and organs were analyzed by flow cytometry 12 h after blockade treatment. (**E**) Bar charts depicting number (#) of NK cells per lung. **F** Bar chart showing the number (#) of differentiated CD11b^high^CD27^low^ NK cells per lung. **G** Representative contour plots of NK cell stages of differentiation

(CD11b^low^CD27^high^, CD11b^high^CD27^high^ and CD11b^high^CD27^low^ cells) in the lung. (**H**) In vitro adhesion assay in which FACS-sorted NK cells were pre-treated with blocking antibodies (αLFA-1, αVLA-4, αLFA-1 + αVLA-4 or isotype control) and incubated with a monolayer of mouse MS1 blood endothelial cells for 30 min. Bar chart depicting frequency of adherent NK cells to MS1 cells. Data are representative for one of 3 independent experiments with n = 3 wells per condition and $n = 9$ for Isotype Ctrl. **I, J** anti-LFA-1, anti-VLA-4, anti-NK1.1 or isotype control antibody were intravenously administered 12 h before injection of PyMT luciferase ($5 \times 10^5$) cells. Tumor cells in the lungs were quantified using IVIS 1 h, 6 h and 24 h post tumor inoculation. **I** Schematic illustration of experimental approach. **J** Kinetics of metastatic load in the lungs measured by IVIS (left) with corresponding representative IVIS measurements ($n = 3$–4 per group). Data are representative for one of 2–3 independent experiments with $n = 3$–5 mice per group. Error bars display means ± SD. Statistical significance was determined by two-way ANOVA with a Sidak's post hoc test. (**E, F, H**) Statistical significance was determined by one way ANOVA with Tukey's multiple comparison test; *$P < 0.05$, **$P < 0.01$, ***$P < 0.001$ and ****$P < 0.0001$. ns, not significant.

(Fig. 3E), corresponding to the highly differentiated CD11b^high^CD27^low^ subset, which constitutes the predominant intravascular population at this time point (Fig. 3F). This effect was not observed for circulating NK cells in the blood (Figure S3C). This was accompanied by an increase of NK cells undergoing late apoptosis, as assessed by Apotracker staining (Fig. S3D, E). We also examined whether the blockade of LFA-1 and VLA-4 affects the interaction between NK and the endothelium directly by performing in vitro adhesion assays. In line with the in vivo findings, inhibition of LFA-1 and VLA-4 led to decreased numbers of NK cells adhering to activated endothelial cells (Fig. 3H).

We next sought to understand whether disrupting the LFA-1 and VLA-4 interactions would have consequences for NK cell-mediated metastatic surveillance. We thus used anti-LFA-1 and anti-VLA-4 blocking antibodies and analyzed metastatic clearance early after tumor cell inoculation using bioluminescent imaging (Fig. 3I). The blockade of either LFA-1 and VLA-4 individually led to a reduced tumor clearance of PyMT-B6 tumor cells (Fig. 3J). Strikingly, the combined inhibition of both LFA-1 and VLA-4 completely prevented tumor clearance and phenocopied a full NK cell depletion (Fig. 3J). Since dual blockade also increased NK cell apoptosis (Fig. S3E), this likely contributes to the observed impairment in tumor clearance. Collectively, these findings underscore a key role for LFA-1 and VLA-4 in maintaining the intravascular subset of differentiated NK cells attached to the pulmonary vasculature, which ensures metastatic clearance.

## Metastatic nodules harbor primarily CD27high NK cells, while differentiated CD11bhigh NK cells remain intravascularly

Given the intravascular localization of pulmonary NK cells during early stages of metastasis, we asked whether these cells would infiltrate the growing metastatic nodules at a later disease stage. We thus investigated the distribution of NK cells in metastatic lungs 15 days after PyMT-B6 tumor cell inoculation. Using flow cytometry, we only observed a small percentage of extravascular NK cells within the whole metastatic lung (Fig. 4A). However, when we manually separated the nodules from the adjacent lung tissue, we found around 40% of total NK cells in the extravascular compartment (Fig.4B, C), which we confirmed by immunofluorescence (Fig. 4D). The amount of extravascular NK cells in the metastatic nodules was further influenced by the tumor type. More immunogenic tumors, such as the colorectal carcinoma MC38, promoted greater NK cell infiltration into the lung metastatic nodules compared to the less immunogenic B16-F10 melanoma (Fig. S4A, B). Interestingly, extravascular NK cells in the metastatic nodules displayed features of CD27^high^ NK cells, with low levels of CD11b, KLRG-1 and Gzmb (Fig. 4E, F). In contrast, NK cells localized within the vasculature of the nodule retain a highly differentiated CD11b^high^ phenotype, more similar to what we observed in the adjacent

healthy lung tissue (Fig. 4E, F). To validate this pattern across models, we used an orthotopic 4T1 mammary tumor and a PyMT model with surgical resection of the primary tumor. In both cases, we observed similar NK cell compartmentalization, with less differentiated CD27^high^ NK cells enriched in the extravascular space, while the more differentiated CD11b^high^ NK cell subset predominated in the vasculature (Figure S4C–J). These findings demonstrate that NK cell localization within metastatic lesions is tightly linked to their maturation state, and that this spatial-maturation relationship is conserved across both experimental and spontaneous metastasis models.

To gain further insights into these cells, we performed scRNA-seq on sorted CD45⁺lin⁻NK1.1⁺NKp46⁺ cells from naive and metastatic lungs, together with corresponding blood (Fig. 4G–J and Fig. S5A, B). We identified two clusters of highly differentiated NK cells characterized by the expression of *Itgam* (CD11b), *Zeb2, Gzma, Gzmb* and *Prf1* (Perforin). Among these, a more terminally differentiated subset (Terminally diff. NK) showed enriched *Cx3cr1* expression (Fig. 4H, I and Fig. S5A, B). A related subset termed Activated NK shared this differentiated signature but additionally expressed effector genes such as *Ifng* and the chemokines *Ccl3* and *Ccl4*[41] (Fig. 4H, I and Fig. S5A, B). Together, these differentiated subsets accounted for the majority of NK cells, consistent with our flow cytometric analysis (Fig. 2A, B). We also detected a less differentiated cell cluster featuring *CD27, Emb, Ccr2, Cd28, Cxcr4, Thy1* and *Sell* (CD62L) (Less diff. NK) (Fig. 4H, I and Fig S5A, B). Whereas naïve lung and blood mostly contained differentiated NK cells, metastatic lungs harbored a large cluster of tumor-associated NK cells (Tumor NK) expressing markers of less differentiated NK cells (*Emb, Cd27, Ccr2, Cd28, Thy1* or *Sell*) (Fig. 4H–J and Fig. S5A, B), confirming our flow cytometric data (Fig. 4E, F). Notably, these tumor-associated NK cells also upregulated genes indicative of tissue-residency and TGF-β imprinting (*Itga1* (CD49a), *Tgfb1, Smad7, Pmepa1* or *Ski*) (Fig. 4G–J and Figure S5-A, B), resembling NK cell populations previously described in primary tumors and liver metastases that exhibit impaired tumor control[42–45].

We next investigated whether these observations are relevant to human cancer by re-analyzing a public scRNA-seq dataset of circulating and tumor-associated NK cells from Non-small cell lung cancer (NSCLC) patients[46]. This analysis revealed six distinct clusters of NK cells (C0-C5) expressing *NCR1* (NKp46), *NCAM1* (CD56) and *FCGR3A* (CD16) (Fig. 4K–M). Intriguingly, two of these clusters (C0 and C1) were predominantly found in the circulation and nearly absent in the tumor (Fig. 4K–M), which instead featured the emergence of cluster (C3) (Fig. 4K–M). The circulating clusters (C0 and C1) were characterized by a differentiated CD56^dim^CD16^pos^ phenotype expressing *CX3CR1, FCGR3A, PRF1, GZMB* and *KLRG1* (Fig. 4M, N). In contrast, the tumor-specific cluster C3 appears less differentiated with features typical of CD56^bright^CD16^neg^ NK cells (*NCAM1, CD2, CD27, SELL, CD69*) (Fig. 4M, N)

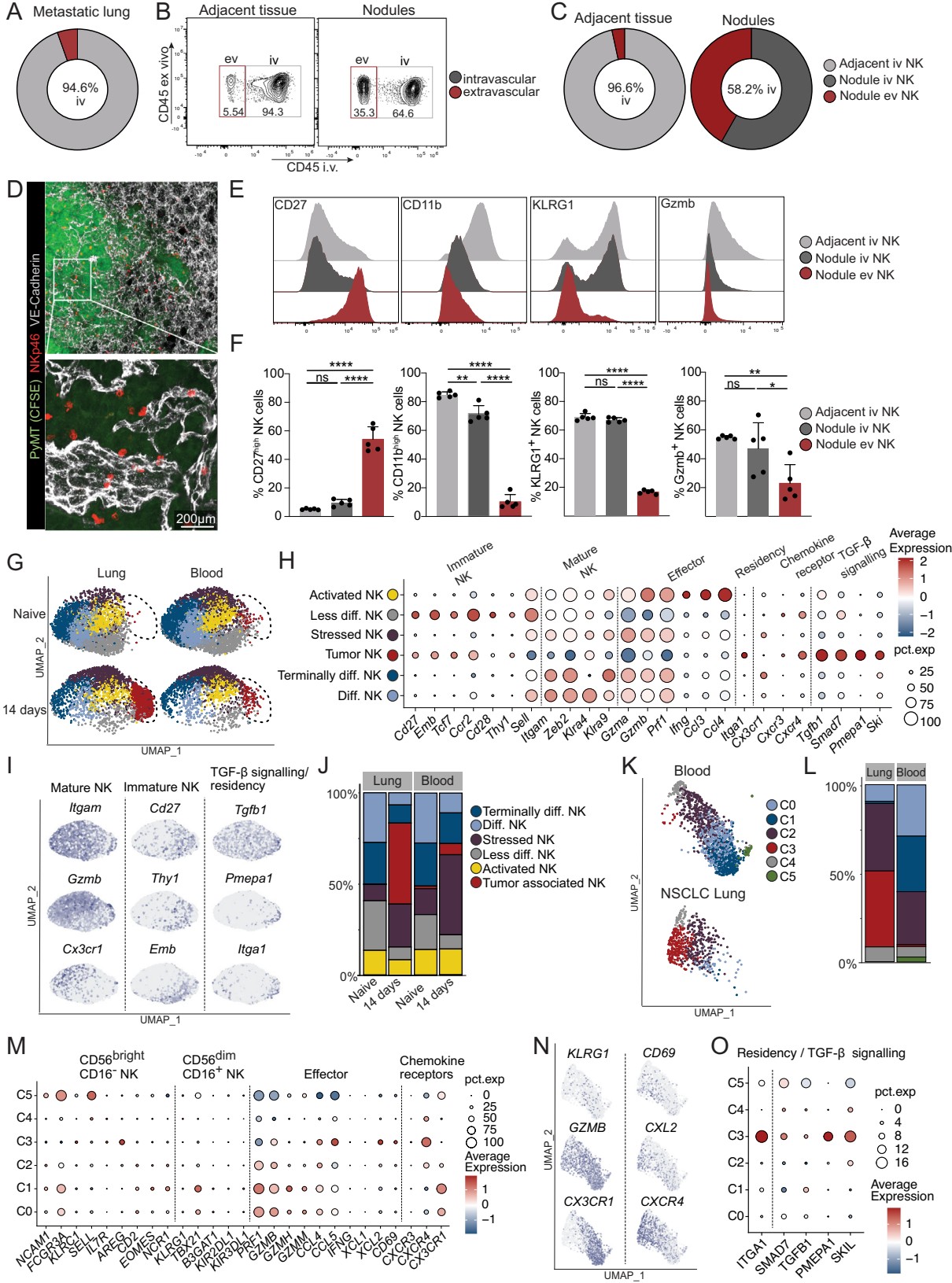

as well as a signature of tissue residency (*CD69* and *ITGA1*) and of TGF-β-induced genes (*SMAD7, TGFB1, PMEP1, SKIL*) (Fig. 4O). Thus, in line with our findings in the murine metastatic model, patients with lung tumors showcase a compartmentalization of NK cells characterized by high frequencies of the more differentiated subsets in the blood, while less differentiated NK cells predominate in the tumor tissue[46].

**Metastatic nodules preferentially attract less differentiated CD27high NK cells while impairing the migration and cytotoxicity of the differentiated CD11bhigh subset**

Based on our observation that tumor-associated NK cells resemble CD27[high] NK cells, we hypothesized that this subset is preferentially recruited to the lung metastatic nodules. To investigate this, we

**Fig. 4 | The metastatic nodules serve as a hub for less differentiated NK cells, while the differentiated subset predominantly localizes intravascularly.** **A** Donut plot indicating the mean percentage of intravascular (light gray) and extravascular (dark red) NK cells in the lung 14 days after PyMT tumor inoculation. **B, C, E, F** Mice were inoculated i.v. with PyMT cells ($5 \times 10^5$). At sacrifice, adjacent lung tissue and tumor nodules were manually separated and analyzed by flow cytometry on day 14 post-inoculation. **B** Representative contour plots showing intravascular and extravascular NK cells in adjacent lung tissue (left) and nodules (right). **C** Donut plot showing the mean percentage of adjacent intravascular, adjacent extravascular (left), nodule intravascular and nodule extravascular (right) NK cells. **D** Immunofluorescence staining of the lung of $Ncr1^{iCre}Ai14^{fl/wt}$ mice injected with CSFE-labeled PyMT cells ($5 \times 10^5$) (green), 14 days before sacrifice with NKp46[+] cells (red) within the vasculature (VE-cadherin/gray). **E** Representative histograms depicting expression of CD27, CD11b, KLRG1 and Gzmb in adjacent intravascular, nodule intravascular and nodule extravascular NK cells. **F** Bar charts showing the frequency of CD27[high], CD11b[high], KLRG1[+] and Gzmb[+] in adjacent intravascular, nodule intravascular and nodule extravascular NK cells. Data (**A, C, D, F**) are representative for one of 3 independent experiments with $n = 4$–5 mice per group. Error bars display means ± SD. Statistical significance was determined by one way ANOVA with Tukey's multiple comparison test; *$P < 0.05$, **$P < 0.01$, ***$P < 0.001$ and ****$P < 0.0001$. ns, not significant. **G–J** Mice were inoculated i.v. with PyMT tumor cells ($5 \times 10^5$) and transcriptome expression analysis (scRNAseq) of sorted NK cells from the lungs and blood was performed on day 14 post-inoculation. Data shown from one experiment with $n = 5$ mice/group. **G** UMAP displaying clustered (based on gene expression) and manually annotated NK cell subsets in lung and blood of naïve mice and 14 days post-inoculation. **H** Dot plot depicting average expression (color intensity) and the percentage (circle size) of cells expressing specific genes across each cluster. **I** Cluster frequency in lung and blood on the indicated time-point. **J** UMAPs displaying the expression of the indicated genes across the NK cell clusters. **K–O** Publicly available scRNA-seq data[46] of NK cells from blood and lung of patients ($n = 7$) with non-small cell lung carcinoma (NSCLC) was re-analyzed. **K** UMAP displaying clustered and manually annotated NK cell subsets in the blood and lung of NSCLC patients. **L** Cluster frequencies in lung and blood. **M** Dot plot depicting the average expression and percentage of cells expressing the indicated genes across each cluster. **N** UMAPs depicting the expression of the indicated genes across the NK cells. **O** Dot plot showing the expression and percentage of cells expressing the residency and TGF-β signaling genes across each cluster.

adoptively transferred CD27[high] and CD11b[high] NK cells from naïve CD45.1 and CD45.2 mice, respectively, at a 1:1 ratio into $Rag2^{-/-}Il2rg^{-/-}$ mice bearing lung metastases (Fig. 5A). Strikingly, at day 1 after the transfer, we found that CD27[high] NK cells infiltrated the metastatic tissue more efficiently than the differentiated CD11b[high] subset, which instead remained more prevalent in the vasculature (Fig. 5B). Importantly, none of the CD27[high] NK cells had differentiated into CD11b[high] NK cells in this time window (Figure S6A). At six days post-transfer, most intravascular CD27[high] NK cells had differentiated towards the CD11b[high] subset, whereas intranodular NK cells showed an impaired differentiation (Fig. 5C, D and Figure S6B). Furthermore, CD27[high]-derived intranodular NK cells upregulated CD49a, indicative of TGF-β imprinting (Fig. 5E). Although less efficient at infiltrating the metastatic nodules, the CD11b[high] NK cells that extravasated in the metastatic lungs underwent a rapid downregulation of KLRG1, CX3CR1 and Gzmb compared to those found intravascularly in metastatic or naïve lungs (Fig. 5F, G and Figure S6C). CD11b[high] NK cells displayed only limited upregulation of CD49a (Figure S6D), consistent with CD27[high] NK cells being more prone to TGF-β imprinting[47].

Since NK cells in both metastatic nodules and human lung tumors display a TGF-β imprinting gene signature, we next investigated how TGF-β signaling on NK cells would influence the infiltration and intranodular phenotype. To investigate this, we transferred either CD27[high] or CD11b[high] NK cells from CD45.2[+] $Ncr1^{Cre}$ $Tgfr2^{fl}$ mice, lacking TGF-β receptor 2 in NKp46[+] group 1 ILCs, together with CD45.1[+] wild-type (WT) NK cells at a 1:1 ratio into metastasis-bearing $Rag2^{-/-}Il2rg^{-/-}$ mice (Fig. 5H). As expected, upregulation of the residency marker CD49a was reduced in nodule-infiltrating, extravascular CD27[high] NK cells from $Ncr1^{Cre}Tgfbr2^{fl}$ mice, compared to control NK cells (Fig. 5I, J). Similarly, the loss of KLRG1, CX3CR1 and Gzmb observed in extravasated CD11b[high] NK cells was partially rescued when this subset was not able to sense TGF-β (Fig. 5K–M). Hence, the release of TGF-β from the metastatic microenvironment redirects CD27[high] NK cell differentiation towards a tumor-associated state, while impairing effector and cytotoxic features of differentiated NK cells.

Finally, we wanted to understand if TGF-β influences the spatial distribution of NK cells, specifically the compartmentalization of less differentiated CD27[high] NK cells infiltrating the tumor and differentiated CD11b[high] NK cells remaining in the vasculature. At day one post-transfer, more CD11b[high] NK cells localized in the vasculature when TGF-β signaling was absent, potentially due to the rescue of CX3CR1 expression (Fig. 5N). Nevertheless, their recruitment to the extravascular compartment of the nodules was unchanged between $Ncr1^{Cre}Tgfbr2^{fl}$ and $Ncr1^{wt}Tgfbr2^{fl}$ control cells (Fig. 5N). Strikingly, CD27[high] NK cells were significantly more abundant in metastatic nodules when they could not sense TGF-β (Fig. 5O). Together, these findings demonstrate that TGF-β plays a critical role in shaping both the functional state and spatial distribution of NK cells in lung metastases.

## Discussion

NK cells are essential in restricting cancer spread to the lung. To better understand this process, we investigated the timing, spatial distribution, and mechanisms by which NK cells influence the metastatic cascade. Our findings reveal the compartmentalized activity of lung NK cells in tumor surveillance. Tumor cells metastasizing to the lung are rapidly eliminated by a subset of highly differentiated CD11b[high] NK cells confined to the pulmonary vasculature. In contrast, at later stages of metastasis development only the CD27[high] NK cell subset can effectively extravasate and enter tumor tissue. Finally, we show that once in the metastatic tissue, CD27[high] NK cells are rapidly paralyzed by TGF-β, and that blocking this signaling pathway unleashes their expansion potential.

The predominance of differentiated NK cells in murine and human lungs has been long reported[12,24,26,48]. Our findings however demonstrate that this subset of NK cells is preferentially recruited to the lung, rather than the lung being a site of NK cell differentiation. The pulmonary circulation represents a vascular system with unique features, as it needs to accommodate the entire cardiac output while maintaining tissue homeostasis[49]. Our data indicate that, when compared to other vascular beds, the pulmonary endothelia express high levels of CX3CL1 at steady-state conditions. The binding of CX3CL1 to its receptor CX3CR1 has been shown to promote NK cell chemotaxis and activation, enhancing their ability to lyse target cells[50,51]. Along the line, we observed a decreased recruitment of differentiated NK cells to the lung when these cells lacked CX3CR1, which is exclusively expressed in this NK cell subset. While the CX3CL1/CX3CR1 axis seems to influence the preferential migration of NK cells to the lung vasculature, the potential contribution of additional factors requires further investigation. We found LFA-1/ICAM-1 and VLA-4/VCAM-1 to be among the strongest interactions that facilitate the crosstalk between NK cells and pulmonary endothelial cells. Although the expression of both ICAM-1 and VCAM-1 is typically upregulated during inflammatory conditions[52], we observed elevated basal levels of both adhesion molecules in the lung vasculature, potentially attributed to continual exposure to environmental threads. Interestingly, the blockade of LFA-1 and VLA-4 under steady-state conditions resulted in reduced numbers of differentiated NK cells associated with the pulmonary vasculature. Thus, beyond the primary role of these integrins in mediating leukocyte arrest on activated endothelium[53], they appear to be crucial

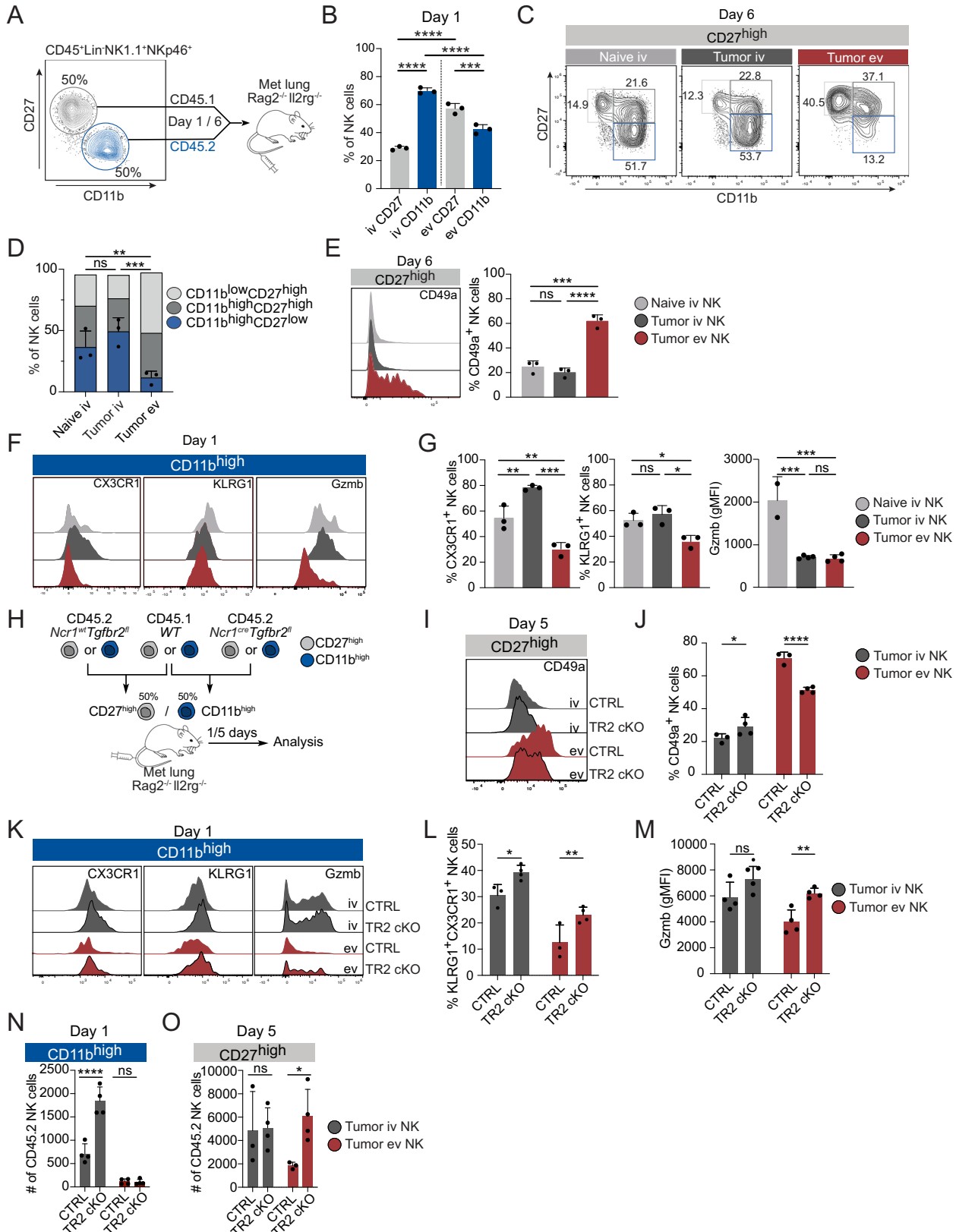

for lung NK cells during homeostasis. Consequently, upon disruption of the LFA-1/ICAM-1 and VLA-4/VCAM-1 axes, NK cells lose their intravascular localization, compromising surveillance of metastasizing tumor cells.

The depletion of NK cells 24 h or 7 days after tumor cell inoculation showed no effect on the metastatic burden in the lung, indicating

that tumor cells extravasated to the lung tissue are no longer controlled by NK cells. We and others have previously shown that NK cells become dysfunctional during metastatic outgrowth, evidenced by their inability to effectively eliminate tumor cells[18,30]. Along the line, nodule-infiltrating NK cells are hindered from undergoing full differentiation and display a rapid downregulation of migration and effector

**Fig. 5 | Metastatic lungs preferentially attract less differentiated NK cells while impairing the migration and cytotoxicity of the differentiated subset. A–H** PyMT cells ($5 \times 10^4$) were injected i.v. into $Rag2^{-/-}Il2rg^{-/-}$ mice. 7 days post-inoculation, FACS-sorted CD45.1[+]CD27[high] (gray) and CD45.2[+]CD11b[high] (blue) NK cells were adoptively transferred at a 1 to 1 ratio and NK cells were analyzed 1 day and 6 days post-transfer. **A** Schematic illustration of experimental approach. **B** Bar charts depicting the frequency of CD45.1[+]CD27[high] and CD45.2[+]CD11b[high] NK cells in the lung comparing intravascular (iv) and extravascular (ev) NK cells at day 1 post-transfer. **D** Representative contour plots and (**D**) bar charts depicting differentiation stages of CD45.1[+]CD27[high]-derived NK cells in naïve intravascular, metastatic tumor intravascular and tumor extravascular lung tissue at day 6 post transfer. **E** Representative histograms and bar charts depicting CD49a expression intravascular and metastatic tumor extravascular from CD27[high]-derived NK cells at day 6 post transfer. **F** Representative histograms and (**G**) bar charts showing CX3CR1, KLRG1 and Gzmb expression in naïve intravascular, metastatic tumor intravascular and tumor extravascular CD11b[high] NK cells at day 1 post-transfer. **H–O** CD45.2 CD27[high] or CD11b[high] NK cells from $Ncr1^{Cre}Tgfr2^{fl}$ (TR2 cKO) mice or $Ncr1^{wt}Tgfbr2^{fl}$ (CTRL) littermate controls, together with the WT CD45.1 counterparts were adoptively transferred at a 1 to 1 ratio into metastatic-bearing $Rag2^{-/-}l2rg^{-/-}$ mice and NK cells were analyzed 1 day and 5 days post-transfer. **H** Schematic illustration of experimental approach. **I** Representative histogram and (**J**) bar charts showing the expression of CD49a on intravascular and extravascular CD45.2[+]CD27[high] NK cells from $Ncr1^{Cre}Tgfr2^{fl}$ (TR2 cKO) or $Ncr1^{wt}Tgfbr2^{fl}$ (CTRL) mice day 5 post-transfer. **K** Representative histograms and bar charts depicting (**L**) CX3CR1, KLRG1 and (**M**) Gzmb expression in intravascular and extravascular CD45.2[+]CD11b[high] NK cells from $Ncr1^{Cre}Tgfr2^{fl}$ (TR2 cKO) or $Ncr1^{wt}Tgfbr2^{fl}$ (CTRL) 1 day post transfer. **N** Bar charts depicting number (#) of intravascular and extravascular CD45.2[+]CD11b[high] NK cells per lung (TR2 cKO or WT) 1 day post transfer. **O** Bar charts depicting number (#) of intravascular and extravascular CD45.2[+]CD27[high] NK cells per lung (TR2 cKO or WT) 5 days post transfer. Data is representative of 3 (**B–G**) or 2 (**I–O**) different experiments with $n = 3$–5. Error bars display means ± SD. Statistical significance was determined by two-way ANOVA with a Sidak's post hoc test (F) or One-way ANOVA with Tukey's multiple comparisons test (**B, D, H, K, M**); *$P < 0.05$, **$P < 0.01$, ***$P < 0.001$ and ****$P < 0.0001$. ns, not significant.

molecules such as CX3CR1 or Gzmb, as described in primary tumors[43,45]. However, we here show that in addition to the tumor-driven suppressive effects, NK cell subsets are inherently different in their capacity to enter and survive in the metastatic tissue, with less differentiated CD27[high] NK cells infiltrating more efficiently than CD11b[high] NK cells. Given the lower effector functions associated with the CD27[high] NK cell subset[24], the observed inability to control metastatic outgrowth might be partially explained by the preferential invasion of CD27[high] cells, rather than solely by the desensitization of CD11b[high] NK cells.

A hallmark of NK cells in both primary tumors and metastasis is their imprinting by TGF-β, which limits antitumor activity[42,44] and occurs shortly after NK cells infiltrate the tumor microenvironment[43,45]. Apart from driving adaptations towards tissue residency, TGF-β is also a key driver of metabolic dysfunction in NK cells, as well as an inhibitor of effector functions and cell surface activating receptors needed[47,54–57]. We here show that TGF-β affects both CD27[high] and CD11b[high] NK cells entering the tumor; however, it is mainly the CD27[high] subset that is driven towards an alternative differentiation program featuring adaptations seen in steady-state NK cell tissue-residency[58,59]. Transfer of NK cells lacking the receptor for TGF-β led to a partial restoration of cytotoxic features among CD11b[high] NK cells and blocked the upregulation of CD49a levels on CD27[high] NK cells. Furthermore, the absence of TGF-β led to enhanced persistence of the CD27[high] NK cell population, suggesting potential therapeutic advantages. Thus, the benefits of TGF-β blockade[32,44] may largely stem from its impact on this tumor-infiltrating CD27[high] NK cell subset. Considering that human tumor-infiltrating NK cells display a similar CD56[bright]CD16[neg] TGF-β imprinted phenotype, NK cell-based therapies might benefit from using TGF-β receptor deficient CD56[bright]CD16[neg] NK cells. However, additional work is still needed to understand the role of other immunosuppressive factors or inhibitory checkpoints that restrict NK cell responses in the metastatic microenvironment.

In conclusion, our study supports a compartmentalized model of NK cell surveillance where distinct populations of NK cells operate within specialized microenvironments to target and control metastatic progression.

## Methods
### Mice
6- to 10-week-old female and male C57BL/6 mice were purchased from Janvier Labs. Ncr1[Cre/wt] mice (B6.Ncr1tm1.1(icre)Viv) were provided by Eric Vivier. Ai14[fl/fl] (B6;129S6-Gt(ROSA)26Sor[tm14(CAG-tdTomato)Hze]/J, stock# 007908), C57BL/6-LY5.1 (CD45.1) mice, Tgfbr2[fl] (B6;129-Tgfbr2[tm1Karl]/J, stock# 012603), Tgfb1[fl] (C57BL/6J-Tgfb1[em2Lutzy]/Mmjax, stock# 065809), Rag2[−/−]Il2ry[−/−] (C;129S4-Rag2[tm1.1Flv]Il2rg[tm1.1Flv]/J, stock #014593), CX₃CR1[GFP] knock-in/knock-out (B6.129P2(Cg)-Cx3cr1[tm1Litt]/J, stock#

005582) were purchased from the Jackson Laboratory. All mice were maintained on a C57BL/6 background and were housed in a pathogen-free environment. Mice were used for experiments at the age of 6–10 weeks. Mice were socially housed with a dark-light cycle of 12 h and 45–65% humidity under specific-pathogen-free conditions according to institutional guidelines in the Laboratory Animal Services Center of the University of Zurich.

All experiments were approved by the Cantonal Veterinary Office of Zurich. Humane endpoints were applied throughout. In the primary mammary fat pad tumor model, mice were euthanized if tumors exceeded 2,000 mm³, if ulceration occurred, if animals displayed severe weight loss or deteriorated body condition, or if pain and distress could not be alleviated by analgesia. In the intravenous lung metastasis model, mice were euthanized if they lost more than 15% of their initial body weight, developed severe respiratory distress or experienced pain that could not be controlled with analgesia. In both models, animals were sacrificed immediately if they reached pre-defined severity limits.

### Tumor cell lines
PyMT cells were a gift from David G DeNardo (Division of Oncology, Washington University School of Medicine). MC38 and B16-F10 melanoma cells were purchased from ATCC. 4T1 cells were provided by Prof. Michael Detmar. Cells were cultured in Dulbecco's Modified Eagle Medium (DMEM, Gibco) supplemented with 10% fetal bovine serum (FBS, ThermoFischer Scientific), 2mM L-glutamine and 2% penicillin/streptomycin (Gibco). Cells were cultured at 37 °C in a humidified atmosphere with 5% $CO_2$.

### Mouse tumor inoculation and treatments
To generate lung metastasis, mice were inoculated intravenously (i.v.) with $1 \times 10^5$ cells in the tail vein. Alternatively, $1 \times 10^5$ cells in 50 µl PBS were injected into the fourth mammary fat pad. Lung metastasis from luciferase-expressing tumor cell lines was quantified using IVIS 200 imaging system (PerkinElmer) 10 min after intraperitoneally (i.p.) injection of 150 mg/kg D-Luciferin (Promega).

For resection of primary tumors, mice received pre-operative analgesia with buprenorphine (0.1 mg/kg, subcutaneously) 30 min prior to surgery. Anesthesia was induced and maintained with 2% isoflurane, and the eyes were protected with Vitamin A cream (Bausch & Lomb). The tumor area was disinfected using 70% ethanol and betadine (Mundi Pharma), followed by the application of xylocaine gel (2%, AstraZeneca) to the skin. Primary tumor were surgically resected using needle-nose forceps, sharp-blunt type scissors and hemostats to cauterize blood vesssels. The surgical wound at the skin surface with non-resorbable sutures (PERMAHAND® Silk Suture, ETHICON). Post-operative analgesia was provided via buprenorphine (0.1 mg/kg, s.c.)

every 4–6 h for 48 h. Additionally, buprenorphine (1 mg/kg) was added to the drinking water.

For depletion of NK cells, 200 µg anti-NK1.1 antibody (clone PK136, BioXcell) was injected i.p. in PBS. Control mice were injected with 200 µg mouse IgG2a isotype control (clone C1.18, BioXcell).

For blockade of LFA-1 and VLA-4, mice were injected with 100 µg anti-LFA-1α antibody (M17/4, BioXcell) and anti-VLA-4 antibody (clone PS/2, BioXcell) i.v. Control mice were injected with 100 µg anti-trinitrophenol (2A3, BioXcell) and anti-keyhole limpet hemocyanin (LTF-2, BioXcell), respectively.

### Tissue processing for single cell suspensions

All mice were euthanized by lethal $CO_2$ inhalation for most experiments. For immunofluorescence staining or experiments involving anti-CD45 intravenous labeling, mice were euthanized by intraperitoneal injection of pentobarbital. In all cases, transcardiac perfusion was performed with ice-cold PBS. For analysis of intravascular leukocytes, anti-CD45-Pacific Blue (Biolegend) (5 µg/mouse) was injected i.v. 5 min prior to sacrifice[33]. Following transcardial PBS perfusion, lungs and spleens were collected in ice-cold PBS. Lungs were minced into pieces using the gentleMACS dissociator (program m_lung_01_02, Miltenyi Biotec) in digestion buffer (HBSS) with calcium/magnesium supplemented with 2% FCS10 mM HEPES, 30 µg ml−1 DNase I and 0.4 mg/ml collagenase IV. Similarly, spleens were cut manually, and the organs were digested in digestion buffer for 30 min at 37 °C with gentle rocking. For the lung, this was followed by further dissociation on the gentleMACS Dissociator (program m_lung_02_01). Organs were filtered through 70 µm cell strainers and washed and incubated in ACK (ammonium-chloride-potassium, Sigma-Aldrich) buffer for 3 min to lyse erythrocytes.

Blood was collected prior to perfusion from the heart and collected in Li-heparin–coated tubes (Sarstedt) and incubated for 15 min in ACK buffer at 4 °C with gentle rocking.

### Flow cytometry

Cells were incubated for 15 min in FcReceptor-blocking buffer (anti-CD16/32 in PBS, Biolegend) before staining. Cells were washed with PBS and incubated in antibody staining for 30 min at 4 °C. In some experiments, biotinylated antibodies for lineage exclusion (CD3ε, CD5, TCRβ, CD19, Ly6G, Ter119) (Table S1) were added to the staining and an additional staining step with a streptavidin-fluorophore conjugate was performed. For intracellular stainings, cells were fixed and permeabilized using Foxp3/transcription factor fixation/permeabilization concentrate and diluent (eBioscience) at 4 °C for 45 min. Cells were washed two times with PermWash (0.01% sodium azide, 0.5% saponin and 2% BSA in PBS) and incubated with antibodies in Permwash at 4 °C overnight. Antibodies were purchased from BioLegend, BD or Thermo Fisher (Table S1). Stained cells were acquired on a Cytek Aurora spectral analyzer (Cytek Biosciences) and analyzed with FlowJo (BD). Dead cells were excluded using Live/Dead fixable staining reagents (LIVE/DEAD Blue or LIVE/DEAD Fixable Near-IR, Thermo Fisher) and doublets were excluded by FSC-A/FSC-H and SSC-A/SSC-H gating.

### Cell sorting

Preparation of single-cell suspensions and subsequent procedures were performed as described above under sterile conditions. Single cells were stained with biotinylated antibodies against lineage (CD3ε, CD5, TCRβ, CD19, Ly6G, Ter119) for 30 min, cells were washed and negatively selected of Lineage+ cells using Streptavidin Nanobeads (BioLegend) for 30 min at 4 °C. Cells were stained with antibodies and afterwards kept in HBSS (without $Mg_2$ and $Cl_2$, Gibco) supplemented with 30% FBS at 4 °C. Sorting was performed using a FACSymphony S6 cell sorter or FACSAria III (BD).

### Adoptive transfers

The cells were sorted using a FACS cell sorter, and gates were set to specifically isolate single-positive $CD27^+CD11b^-$ and $CD11b^+CD27^-$ NK cells. To confirm the purity of the sorted populations, we performed a re-analysis directly after sorting. Sorted cells were washed, counted and resuspended in sterile PBS and injected i.v. in the tail vein of $Rag2^{-/-}Il2r^{-/-}$ recipients. For the transfers 200,000 NK cells were injected. NK cells were analyzed using flow cytometry 1, 3, 5 or 6 days after transfer.

### In vitro adhesion assay

Mouse MS1 blood endothelial cells were maintained on culture dishes coated with collagen type-I (Advanced Biomatrix) and fibronectin (Millipore) (both at 10 µg/ml) in Dulbecco's Modified Eagle Medium (DMEM, Gibco) supplemented with 5% FBS. Prior to the experiment, 25,000 MS1 cells / well were seeded on coated 96-well plates. On day 2 after seeding, cells were stimulated overnight with TNFα (Peprotech, 10 µg/ml). The next day, the medium was removed, and 50,000 FACS-sorted NK cells pretreated with blocking antibodies (10 µg/ml of anti-LFA-1 and anti-VLA-4 or isotype control) were added to the MS1 monolayer and incubated on a gyral shaker for 30 min. Subsequently, wells were washed twice with PBS, before trypsinization and counting of the adhered NK cells using a Cytoflex S instrument (Beckmann Colter).

### Immunofluorescence

Mice were euthanized by lethal i.p. injection of pentobarbital and transcardiac perfusions were performed with ice-cold PBS. The tracheae were exposed and lungs were instilled intratracheally with 1 ml of pre-warmed 2% low-melting agarose. Lungs were incubated in cold 4% PFA for 2 h at 4 °C and separate lung lobes were embedded in 2% low-melting agarose. Samples were sectioned using the Leica VT1200 S vibratome to generate 100 to 200 µm thick slices. Slides were washed twice with PBS for 5 min and were incubated in blocking buffer consisting of 5% donkey/goat serum (BioConcept or Thermo Fisher), 0.2% BSA (Sigma Aldrich), 0.3% Triton-X100 (Sigma Aldrich) for 1 h at RT on a gyral shaker. The slices were then incubated overnight at 4 °C with primary antibodies (Table S2) in blocking buffer. Slides were then washed six times with PBS (10 min) at RT on a gyral shaker and stained with secondary antibodies (Table S2) at 4 °C. Slides were washed three times with PBS (10 min) and stained with DAPI for 10 min at 4 °C. Finally, slides were washed three times with PBS (10 min) and mounted. Images were taken on the Leica SP8 Falcon and processed with Imaris.

### Single-cell RNA sequencing

Mice were i.v. inoculated with $1 \times 10^5$ PyMT cells 14 days and 6 h before they were sacrificed. At the endpoint, lungs and blood were prepared as described above. Lung and blood immune cells were distinguished by labeling with unique hastags Totalseq-B0304 and B0306 (Biolegend). Live NK cells from lung and blood were sorted into PBS containing 2% FBS. Viability of the sorted cells was evaluated with trypan blue and was over 95% for all samples. Sorted cells were loaded into 10x Genomics Chromium in parallel. Libraries were prepared following manufacturer's protocol (Chromium Next GEM Single Cell 3′ Reagent Kits v3.1 protocol) and sequenced on an Illumina NovaSeq sequencer according to 10X Genomics recommendations to a depth of around 50,000 reads per cell. Initial processing was done using Cell Ranger (v7.0.1) mkfastq and count (reads were aligned to GENCODE reference build GRCm38.p6 Release M23). Starting from the filtered gene-cell count matrices produced by CellRranger's built-in cell calling algorithms, we proceeded with Seurat v5 workflow[60].

### Downstream analysis of single-cell RNA sequencing

Analysis, including quality control (QC), processing, graph-based clustering, visualizations, and differential gene expression analyses of the scRNA-seq data were performed in R using the Seurat pipeline

(version 5.0)[60]. All samples were merged and processed together. Firstly, potential low-quality cells with number of features below 800 and above 5000, or with more than 6% mitochondrial genes were filtered out using the QC parameters of nFeature_RNA and percent.mt. Global count normalization was performed, followed by selection of variable features (n = 2000), linear transformation using ScaleData, and dimensionality reduction (PCA). Data of lung and blood samples were integrated using FastMNN[61]. Graph-based clustering and dimensionality reduction were performed using the first 30 principal components and a resolution value of 0.8. NK cell cluster assignment were based on cluster-specific genes identified with the FindAllMarkers function (only.pos = TRUE, min.pct = 0.25, logfc.threshold = 0.25). For murine endothelial cells, the dataset of[36] was used.

## Statistics and reproducibility
All experiments were performed using randomly assigned, sex-matched mice without blinding of investigators. No statistical methods were used to determine sample sizes. All data points reflect biological replicates. The results are presented as mean ± SD unless otherwise specified. Statistical significance was determined using Graphpad Prism (GraphPad Software Inc.). Statistical significance was calculated with one-way analysis of variance (ANOVA) with Tukey's multiple comparison test or with a two-way ANOVA with Sidak's posthoc test: P < 0.05 was considered significant (*P < 0.05, **P < 0.01, ***P < 0.001, and ****P < 0.0001; ns, not significant). For all immunofluorescence images shown in panels 1 h, 1i, 3 d, and 4 d, experiments were independently repeated at least 2–3 times to ensure reproducibility of the images.

## Reporting summary
Further information on research design is available in the Nature Portfolio Reporting Summary linked to this article.

## Data availability
The sequencing data generated in this study have been deposited in the Gene Expression Omnibus (GEO) under accession number GSE301222. All other data supporting the findings of this study are either included in the article or are available from the corresponding author upon reasonable request. Source data are provided with this paper.

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

## Acknowledgements

We thank E. Vivier (Aix Marseille University, CNRS, INSERM, CIML) for providing mice and the Cytometry Core Facility (University of Zurich), the Institute of Laboratory Animal Research (University of Zurich), the Center for Microscopy and Image Analysis (University of Zurich) and the Functional Genomics Center Zurich (University of Zurich) for experimental support. We would also like to thank P. Zwicky (Weizmann Institute), F. Ingelfinger (Weizmann Institute), P. Flüchter (University of Zurich), and C. Ulutekin (University of Zurich) for intellectual input and technical support. This work was supported by grants from the Swiss National Science Foundation (PR00P3_179775) to S.T., the Swiss Cancer Research Foundation (KFS-5420-02-2021) to S.T., the Sassella Foundation to C.S. and S.T., the Vontobel Foundation to S.T., the Novartis Foundation to S.T., the Olga Mayenfisch Foundation to S.T. and the Wilhelm Sander Foundation to S.T. For the publication fee we acknowledge financial support by Heidelberg University.

## Author contributions

M.V. designed, conducted, and analyzed the experiments. C.S., N.C., M.M., G.L., A.F., S.D., C.M., L.R., P.H., L.D., T.W. and L.C.D. performed selected experiments and provided technical expertise. T.W., L.C.D. and B.B. provided intellectual input and conceptual advice. S.T. supervised the study and wrote the paper together with M.V. and C.S., with input of all coauthors.

## Funding

## Competing interests

The authors declare no competing interests.
