## [Transparent Peer Review file · Nature Communications]

Tissue localization of Natural Killer cells dictates surveillance of lung metastasis

Corresponding Author: Professor Sonia Tugues

Version 0:

Reviewer comments:

Reviewer #1

(Remarks to the Author)

Vermeer, et. al. has detailed a nuanced and spatially distinct NK cell immunosurveillance of experimental lung metastasis. This work is wonderfully fascinating and well executed. This has been a long-standing question in the field and to see thorough technical exploration is a joy. Given the robust experimental procedures and ease of procuring additional data the following suggestions should be easily implemented.

Major Comments:

Unfortunately, the use of tail vein injected cancer cells does not mimic the actual metastatic cascade. There are numerous methods to augment epithelial-like cancer cells to be more mesenchymal-like which would better mimic "true" metastasis. I understand this is mildly controversial in the EMT/metastasis field. However, an injection of cultured cancer cells into the vein is an even further departure.

The cancer cells utilized can be used in orthotopic models, PyMT can be easily injected into the mammary fat pad and the lung assessed for metastasis. Further, LLC can be implanted subcutaneously, and a surgical resection can be performed to allow for lung metastasis analysis and NK cell targeting as described.

The above experiments would enhance the impact of the work greatly as the cancer cells would have adjusted to the microenvironments and putatively undergone the full primary tumor cell to metastasizing cell to secondary tumor growth cascade. This transition period in the blood is highly dynamic and it is known that platelets will bind and provide both a physical barrier and aid in immunosuppressive cytokine secretions to block cytotoxic cells. Most prevalently through TGF- β . Based on the data provided, these new experiments may alter the interpretation of which NK cell subsets are responsible for killing the metastatic cell compared to a transitioning cell back to an epithelial-like mass.

Minor Comments

Additionally, epithelial cells injected may die due to anoikis and confound some of the killing kinetics observed. This is most exemplified by the data presented in supplemental figure 1B. Both conditions see an equal drop in radiance indicating this loss of signal and by proxy cancer cells. Given the speed and variance in signal after the 1 hour mark it would be surprising to attribute NK mediated cytotoxicity in that first hour or even hours.

MFI is used predominantly throughout, and the use of the geometric MFI would be better. There can be skewing in some of the flowcytometry plots and the gMFI would correct for that.

Reviewer #2

(Remarks to the Author)

Vermeer et al study the ability of various subsets to populate healthy and tumour tissue in the mouse lung. The authors conclude that the pulmonary vasculature recruit and maintain cytotoxic CD11b+ NK cells through LFA-1 and VLA-4. These

NK cells are assumed to kill tumor cells within the vasculature, but not to be able to extravasate into the lung and tumour tissue. CD27^{high} NK cells on the other hand are capable of infiltrating metastatic nodules. TGF- β is shown to influence NK cell phenotype and cytotoxic features in the TME. I think that the overall conclusions are warranted even if some experimental findings are over-interpreted leading to conclusions that may also have alternative explanations. My comments and questions are listed below.

1. About the experimental model system. Are metastases formed in other organs than lung? When the lungs are prepared for analysis are all blood cells retained in the capillaries or only those that stick to the vessel walls, i.e. could the analyzed NK cells represent a subgroup of all NK cells in the blood of the lung?
2. Row 124. How do the authors know that elimination of tumors cells take place upon arrival in the lung and not in other places in the blood?
3. Row 155-156. CD27^{high} and CD11b^{high} were transferred in equal proportion to a recipient mouse. How were these cells isolated and prepared to avoid double positive cells (CD27+CD11b)?
4. Row 221: "Interestingly, the highly differentiated subset of CD11b^{high}CD27^{low} NK cells decreased the most upon LFA-1 and VLA-4 blockade (Fig 3F-G)" I'm not sure what data supports this conclusion?
5. Fig S3E shows that NK cells enter apoptosis upon blocking with anti-LFA-1 and anti-VLA-4 together. This is important for the interpretation of Fig 3J since the NK cell death could be contributing to the strong effect observed for double blocking. This should be mentioned in the summarizing section (row 233-235).
6. There is quite extensive literature showing that expression of adhesion molecules like CD11b, LFA-1 and VLA-4 are important for trans-endothelial migration (TEM). (See for example: <https://pubmed.ncbi.nlm.nih.gov/9645378/>) Here the authors find that expression of these molecules prevent TEM. This should be discussed.
7. It is unclear how the 6 subpopulations in Fig 4G are defined and named.
8. "Nevertheless, CD27^{high} NK cells were still preferentially recruited to the metastatic nodule, showing that this occurs independently of TGF- β ." What are the data supporting this?

Minor comments

Row 61. What is "n.d." in the reference parenthesis?

Fig 3J. The colors are hard to distinguish from each other. Maybe change colors or alter the symbols as well.

Fig 4 C-D are mentioned before Fig 4B. Could easily be changed.

Row 1026. bar charts should be bar charts

Reviewer #3

(Remarks to the Author)

This work provides a good evaluation of the compartmentalization of NK cells involved in the tumor surveillance. The paper highlights a very important aspect of the interaction between NK and tumor cells with the final aim of the characterization of the different morphological and molecular aspects underlying this interaction. The focus of this work investigated an important aspect to better understand the NK activity in the lung metastasis. The data showed is very strong because authors apply different experimental approaches with high level of analyses. The work meets the expected standards of the field with enough detailed methods that allow to reproduce the experimental settings.

Please provide:

- The same magnification in Fig 1 H/I.
- Fig 3 J what happen after 24 hours? Statistical differences are confirmed?
- Provide a reference for "in line with our findings ...patients with lung tumors ..." line 295

Version 1:

Reviewer comments:

Reviewer #1

(Remarks to the Author)

The authors have answered my questions and performed more experimentation to support their claims.

Reviewer #2

(Remarks to the Author)

The authors have addressed my comments in a satisfactory way.

(The new text in rows 432 and 434 seems to be missing from the PDF version I downloaded but it is present in the corresponding DOC file, which I guess is more important.)

Reviewer #3

(Remarks to the Author)

The authors performed a high-quality work, providing comprehensive responses to the requests. The answers provided were clear, precise, and well articulated.

Reviewer 1

Vermeer, et. al. has detailed a nuanced and spatially distinct NK cell immunosurveillance of experimental lung metastasis. This work is wonderfully fascinating and well executed. This has been a long-standing question in the field and to see thorough technical exploration is a joy. Given the robust experimental procedures and ease of procuring additional data the following suggestions should be easily implemented.

We thank R1 for the very positive feedback of our work.

Major Comments:

Unfortunately, the use of tail vein injected cancer cells does not mimic the actual metastatic cascade. There are numerous methods to augment epithelial-like cancer cells to be more mesenchymal-like which would better mimic “true” metastasis. I understand this is mildly controversial in the EMT/metastasis field. However, an injection of cultured cancer cells into the vein is an even further departure. The cancer cells utilized can be used in orthotopic models, PyMT can be easily injected into the mammary fat pad and the lung assessed for metastasis. Further, LLC can be implanted subcutaneously, and a surgical resection can be performed to allow for lung metastasis analysis and NK cell targeting as described. The above experiments would enhance the impact of the work greatly as the cancer cells would have adjusted to the microenvironments and putatively undergone the full primary tumor cell to metastasizing cell to secondary tumor growth cascade. This transition period in the blood is highly dynamic and it is known that platelets will bind and provide both a physical barrier and aid in immunosuppressive cytokine secretions to block cytotoxic cells. Most prevalently through TGF- β . Based on the data provided, these new experiments may alter the interpretation of which NK cell subsets are responsible for killing the metastatic cell compared to a transitioning cell back to an epithelial-like mass.

We acknowledge Reviewer 1’s concern regarding the use of tail vein-injected tumor cells. We selected this model because it allows for controlled and synchronous analysis of early phases of lung colonization. However, to address this limitation, we have complemented our study with two additional models: an orthotopic 4T1 mammary fat pad model and a PyMT model with surgical resection of the primary tumor. These models allow for spontaneous tumor cell dissemination and incorporate key steps such as intravasation, circulation, and colonization of distant organs. Using intravascular labeling in these models, we investigated the distribution of NK cells in metastatic lungs. In both cases, we observed similar NK cell compartmentalization, with less differentiated CD27^{high} NK cells enriched in the extravascular space, while the more differentiated CD11b^{high} NK cell subset predominated in the vasculature (Fig. S4C-J). Consistent with the tail vein injection model, extravascular NK cells displayed a CD27^{high} phenotype with reduced expression of KLRG1 and Granzyme B (Fig. S4C-

J), while intravascular NK cells retained a highly differentiated phenotype. These findings support a consistent link between NK cell localization and maturation status across different models of metastasis.

In the PyMT model, we additionally depleted NK cells either before or after tumor resection and observed an increased metastatic burden only when depletion occurred before resection, consistent with our observations in the i.v. model, where NK cells act early to control metastatic seeding (Suppl. Figure 1A–D).

Minor Comments

Additionally, epithelial cells injected may die due to anoikis and confound some of the killing kinetics observed. This is most exemplified by the data presented in supplemental figure 1B. Both conditions see an equal drop in radiance indicating this loss of signal and by proxy cancer cells. Given the speed and variance in signal after the 1 hour mark it would be surprising to attribute NK mediated cytotoxicity in that first hour or even hours.

We agree that anoikis-induced cell death may contribute to the initial drop in radiance signal shortly after intravenous injection of tumor cells. However, NK cell depletion clearly alters the kinetics of this signal loss, suggesting that NK cells actively contribute to early tumor clearance. While we cannot exclude a contribution of anoikis, the difference in radiance signal between control and NK-depleted mice within the first few hours after tumor cell inoculation supports the notion that NK cells mediate rapid cytotoxic activity in this model.

MFI is used predominantly throughout, and the use of the geometric MFI would be better. There can be skewing in some of the flowcytometry plots and the gMFI would correct for that.

We appreciate R1's comment regarding the use of MFI and the recommendation to use geometric MFI (gMFI). We would like to indicate that gMFI was used throughout the manuscript but was referred to as "MFI" in the text and figure legends. We have corrected this for clarity.

Reviewer 2

Vermeer et al study the ability of various subsets to populate healthy and tumour tissue in the mouse lung. The authors conclude that the pulmonary vasculature recruit and maintain cytotoxic CD11b+ NK cells through LFA-1 and VLA-4. These NK cells are assumed to kill tumor cells within the vasculature, but not to be able to extravasate into the lung and tumour tissue. CD27^{high} NK cells on the other hand are capable of infiltrating metastatic nodules. TGF- β is shown to influence NK cell phenotype and cytotoxic features in the TME. I think that the overall conclusions are warranted even

if some experimental findings are over-interpreted leading to conclusions that may also have alternative explanations. My comments and questions are listed below.

1. About the experimental model system. Are metastases formed in other organs than lung? When the lungs are prepared for analysis are all blood cells retained in the capillaries or only those that stick to the vessel walls, i.e. could the analyzed NK cells represent a subgroup of all NK cells in the blood of the lung?

We thank reviewer 2 for the thoughtful question regarding our experimental model of metastasis. In this model, metastases form almost exclusively in the lungs. This is consistent across experiments and supported by whole-body IVIS imaging (see representative image below), which show a strong and localized signal in the thoracic region and absence of signal in other organs. To analyze lung-associated immune cells, we perfuse the lungs thoroughly prior to tissue processing. We therefore assume that the majority of NK cells retained in the lungs at the time of analysis are those adhering to the vasculature or inside the parenchyma, rather than freely circulating cells.

The experimental concerns have been clarified and incorporated into the manuscript.

PBP FIG 1. Representative whole body In Vivo Imaging Systems (IVIS) image at 8 days post-i.v. injection of PyMT luciferase cells.

2. Row 124. How do the authors know that elimination of tumor cells take place upon arrival in the lung and not in other places in the blood?

We thank Reviewer 2 for this important point. While we cannot fully exclude the possibility that some tumor cell elimination occurs in circulation, several observations suggest that NK cell-mediated killing primarily takes place in the lung vasculature. IVIS imaging shows that tumor cells rapidly localize to the lungs within minutes after intravenous injection, indicating that they are quickly retained in the lung vasculature. Moreover, we observed no significant difference in metastatic burden between NK cell-depleted and control mice at 1 hour post-injection, whereas differences became apparent at later time points (from around 3 hours onward) (Figure 1D). These findings are consistent with the idea that NK cells act after tumor cells have lodged in the lung, rather than during circulation.

Row 155-156. CD27^{high} and CD11b^{high} were transferred in equal proportion to a recipient mouse. How were these cells isolated and prepared to avoid double positive cells (CD27⁺CD11b⁺)?

We thank Reviewer 2 for the comment regarding the transfer of CD27⁺ and CD11b⁺ NK cell subsets. The cells were sorted using a FACS cell sorter, and gates were set to isolate single-positive CD27⁺CD11b^{low} and CD11b⁺CD27^{low} NK cells. To confirm the purity of the sorted populations, we performed a re-analysis directly after sorting. A representative FACS plot is shown below, demonstrating that the sorted populations contained minimal amounts of double-positive cells. We have clarified this important point in the Material and Methods section.

PBP FIG 2. Representative FACS plot of sorted CD27^{high} and CD11b^{high} NK cells, which were counted and mixed at a 1:1 ratio prior to injection.

3. Row 221: “Interestingly, the highly differentiated subset of CD11b^{high}CD27^{low} NK cells decreased the most upon LFA-1 and VLA-4 blockade (Fig 3F-G)” I’m not sure what data supports this conclusion?

We thank Reviewer 2 for pointing this out and agree that the original sentence lacked precision. We have revised the text and substituted it for the following:

“Inhibition of LFA-1 and VLA-4 led to decreased numbers of intravascular lung NK cells (Fig. 3E), corresponding to the highly differentiated CD11b^{high}CD27^{low} subset, which constitutes the predominant intravascular population at this time point (Fig. 3F). This effect was not observed for circulating NK cells in the blood (Fig. S3C)”.

5. Fig S3E shows that NK cells enter apoptosis upon blocking with anti-LFA-1 and anti-VLA-4 together. This is important for the interpretation of Fig 3J since the NK cell death could be contributing to the strong effect observed for double blocking. This should be mentioned in the summarizing section (row 233-235).

We thank the reviewer for this suggestion and have incorporated this point in the summarizing section as following: “Since dual blockade also increased NK cell

apoptosis (Fig. S3E), this likely contributes to observed impairment in tumor clearance”.

6. There is quite extensive literature showing that expression of adhesion molecules like CD11b, LFA-1 and VLA-4 are important for trans-endothelial migration (TEM). (See for example: <https://pubmed.ncbi.nlm.nih.gov/9645378/>) Here the authors find that expression of these molecules prevent TEM. This should be discussed.

We thank R2 for this relevant reference. Adhesion molecules such as CD11b, LFA-1, and VLA-4 promote transendothelial migration (TEM) by mediating firm adhesion and facilitating extravasation. Our findings, showing reduced NK cell accumulation in the lung upon LFA-1 and VLA-4 blockade, go in the same direction and likely result from impaired adhesion, as confirmed by our *in vitro* adhesion assays.

7. It is unclear how the 6 subpopulations in Fig 4G are defined and named.

We have clarified this in the text and referred to the names given in the main figure to each of the populations. “We identified two clusters of highly differentiated NK cells characterized by the expression of *Ilgam* (CD11b), *Zeb2*, *Gzma*, *Gzmb* and *Prf1* (Perforin). Among these, a more terminally differentiated subset (“Terminally diff. NK”) showed enriched *Cx3cr1* expression (Fig. 4H-I and Fig S5A-B). A related subset “Activated NK” shared this differentiated signature but additionally expressed effector genes such as *Ifng* and the chemokines *Ccl3* and *Ccl4* (Ni et al., 2020) (Fig. 4H-I and Fig S5A-B). Together, these differentiated subsets accounted for the majority of NK cells, consistent with our flow cytometric analysis (Fig. 2A-B). We also detected a less differentiated cell cluster featuring *CD27*, *Emb*, *Ccr2*, *Cd28*, *Cxcr4*, *Thy1* and *Sell* (CD62L) (“Less diff. NK”) (Fig. 4H-I and Fig S5A-B). Whereas naïve lung and blood mostly contained differentiated NK cells, metastatic lungs harboured a large cluster of tumor-associated NK cells (“Tumor NK”) expressing markers of less differentiated NK cells (*Emb*, *Cd27*, *Ccr2*, *Cd28*, *Thy1* or *Sell*) (Fig. 4H-J and Fig S5A-B), confirming our flow cytometric data (Fig. 4E-F). Notably, these tumor-associated NK cells also upregulated genes indicative of tissue-residency and TGF- β imprinting (*Itga1* (CD49a), *Tgfb1*, *Smad7*, *Pmepa1* or *Ski*) (Fig. 4G-J and Fig. S5-A-B), resembling NK cell populations previously described in primary tumors and liver metastases that exhibit impaired tumor control (Dean et al., 2024; Ducimetière et al., 2021; Gao & Smyth, 2017; Kirschenbaum et al., 2024).”

8. “Nevertheless, CD27^{high} NK cells were still preferentially recruited to the metastatic nodule, showing that this occurs independently of TGF-b.” What are the data supporting this?

We agree this sentence could be better phrased. What we intended to convey is that CD27^{high} NK cells still accumulate and even persist better in the metastatic nodules when TGF- β signaling is absent. We have rephrased it in the discussion section with

the following: “Furthermore, the absence of TGF- β led to enhanced persistence of the CD27^{high} NK cell population, suggesting potential therapeutic advantages. Thus, the benefits of TGF- β blockade (Ducimetière et al., 2021; Gao & Smyth, 2017) may largely stem from its impact on this tumor-infiltrating CD27^{high} NK cell subset”.

Minor comments

Row 61. What is “n.d.” in the reference parenthesis?

This was a mistake and has been changed.

Fig 3J. The colors are hard to distinguish from each other. Maybe change colors or alter the symbols as well.

For better clarity, we have changed the symbols of this graph.

Fig 4 C-D are mentioned before Fig 4B. Could easily be changed.

We agree and we have changed this accordingly.

Row 1026. bar charts should be bar charts

Thanks, we have changed this.

Reviewer 3

This work provides a good evaluation of the compartmentalization of NK cells involved in the tumor surveillance. The paper highlights a very important aspect of the interaction between NK and tumor cells with the final aim of the characterization of the different morphological and molecular aspects underlying this interaction. The focus of this work investigated an important aspect to better understand the NK activity in the lung metastasis. The data showed is very strong because authors apply different experimental approaches with high level of analyses. The work meets the expected standards of the field with enough detailed methods that allow to reproduce the experimental settings.

We thank R3 for the feedback and positive assessment of our work.

Please provide:

- The same magnification in Fig 1 H/I.

We provided the same magnification in Fig 1 H/I.

- Fig 3 J what happen after 24 hours? Statistical differences are confirmed?

When we extended our analysis to a later time point (72 hours post-inoculation), we found that although some tumor cells had already contracted and been eliminated, significant differences in tumor burden between the control group and mice treated with either anti-LFA-1 or anti-VLA-4 persisted (see Figure below).

PBP FIG 3. (A) Kinetics of metastatic load in the lungs measured by IVIS 1, 6 and 72 hrs after tumor inoculation.

- Provide a reference for “in line with our findings ...patients with lung tumors ...” line 295

The reference has been added accordingly.